# Multilevel factors drive child exposure to enteric pathogens in animal feces: A qualitative study in northwestern coastal Ecuador

April M. Ballard[1,2], Betty Corozo Angulo[3], Nicholas Laramee[4], Jayden Pace Gallagher[2], Regine Haardörfer[5], Matthew C. Freeman[2], James Trostle[6], Joseph N. S. Eisenberg[7], Gwenyth O. Lee[8], Karen Levy[9]*, Bethany A. Caruso[2,4,5]

1 Department of Population Health Sciences, Georgia State University School of Public Health, Atlanta, Georgia, United States of America, 2 Gangarosa Department of Environmental Health, Emory University Rollins School of Public Health, Atlanta, Georgia, United States of America, 3 Universidad Técnica Luis Vargas Torres de Esmeraldas, Esmeraldas, Ecuador, 4 Hubert Department of Global Health, Emory University Rollins School of Public Health, Atlanta, Georgia, United States of America, 5 Department of Behavioral, Social, and Health Education Sciences, Emory University Rollins School of Public Health, Atlanta, Georgia, United States of America, 6 Department of Anthropology, Trinity College, Hartford, Connecticut, United States of America, 7 Department of Epidemiology, University of Michigan School of Public Health, Ann Arbor, Michigan, United States of America, 8 Rutgers Global Health Institute and Department of Biostatistics and Epidemiology, Rutgers School of Public Health, Piscataway, New Jersey, United States of America, 9 Department of Environmental and Occupational Health Sciences, University of Washington School of Public Health, Seattle, Washington, United States of America

* klevyx@uw.edu

**Data Availability Statement:** All relevant data are within the paper, which constitutes the minimal

## Abstract

Exposure to animal feces and associated enteric pathogens poses significant risks to child health. However, public health strategies to mitigate enteric infections among children largely aim to reduce exposure to human feces, overlooking transmission pathways related to animal feces. In this study we examine if and how children are exposed to enteric pathogens in animal feces in northwestern coastal Ecuador. We conducted qualitative interviews with mothers of children aged 10–18 months that owned ($n = 32$) and did not own ($n = 26$) animals in urban and rural communities. Using thematic analysis, we identified community, household, and child behavioral factors that influence exposure. We also compared child exposure by household animal ownership. Our findings revealed myriad opportunities for young children to be exposed to enteric pathogens in many locations and from multiple animal sources, regardless of household animal ownership. Animal feces management practices (AFM) used by mothers, such as rinsing feces into ditches and throwing feces into surrounding areas, may increase environmental contamination outside their homes and in their communities. Unsafe AFM practices were similar to unsafe child feces management practices reported in other studies, including practices related to defecation location, feces removal and disposal, environmental contamination cleaning, and handwashing. Findings suggest that animal feces may contaminate the environment along similar pathways as human feces. Identification and incorporation of safe AFM practices, similar to those developed for child feces management, would 1) mitigate child exposure to enteric pathogens by

dataset for the study. For more information, contact April Ballard (aballard11@gsu.edu).

**Funding:** This work is supported by the National Institutes of Health (R01AI137679 to JNSE and KL), which provided financial support to AMB, BCA, MCF, GOL, and BAC. The funders had no role in study design, data collection and analysis, decision to publish, or preparation of the manuscript. The content is solely the responsibility of the authors and does not necessarily represent the official views of the National Institutes of Health.

**Competing interests:** The authors have declared that no competing interests exist.

reducing animal feces contamination in domestic and public spaces; and 2) enable an integrated approach to address enteric pathogen exposure pathways related to animal and child feces.

## Introduction

Exposure to enteric pathogens during childhood is associated with substantial disease burden. Enteric infections and diarrheal diseases are the fifth leading cause of death in children under age five globally [1–5]. Persistent exposure to enteric pathogens during childhood can result in recurrent infections and lifelong consequences, such as deficits in growth and cognitive development [6–12]. Children in low- and middle-income countries (LMICs) bear the greatest burden of enteric disease due to pervasive fecal contamination of domestic environments resulting from inequities in access to improved water, sanitation, and hygiene (WASH) [13, 14].

Interrupting the principal fecal-oral transmission pathways is critical to preventing enteric infections and related adverse health outcomes. Transmission principally occurs when feces from an infected host contaminate fluids, food, fomites, fingers, fields, and flies, followed by human exposure to the contamination through ingestion. This process is often visually depicted as the 'F-diagram' [15–19]. The provision of WASH services is a well-established public health strategy to prevent transmission of enteric pathogens, typically by targeting exposure to human feces [14, 16, 20, 21]. However, transmission of enteric pathogens from animal feces has been overlooked in most WASH programming to date [19, 22–24], despite the fact that animals produce approximately four times as much feces as humans [25] and many enteric pathogens capable of infecting humans are found in animal feces (e.g., *Campylobacter* spp., *Cryptosporidium* spp., enteropathogenic *E. coli*) [26].

Understanding the upstream causes of environmental fecal contamination and child exposure to enteric pathogens in animal feces will be critical to the development of effective mitigation strategies to integrate into WASH programming. Various community, household, and child practices and behaviors can play a key role in exacerbating or mitigating exposure to enteric pathogens in animal feces. Animal husbandry and feces management practices, which are determined by diverse household and community factors, can increase contamination of the environment [18, 27]. Children are then exposed through their interactions with animals, the environment and objects [18]. Current evidence is minimal and insufficient for determining a generalizable set of behaviors that influence zoonotic exposures [18, 19, 24, 28]. However, community- and household-level factors related to animal husbandry and feces management [18, 23, 24, 27, 29]. may be root causes of exposure. For example, animal feces may be abundant throughout the domestic environment, regardless of household-level animal ownership, because letting animals roam freely to forage for food is a community norm that is perceived as beneficial to animals and reduces the financial burden of animal feed [27, 30–33].

To address these key knowledge gaps, we qualitatively characterize exposure to enteric pathogens in animal feces among children in northwestern coastal Ecuador, a high enteric pathogen transmission setting. Previous studies have estimated the two-week prevalence of diarrhea among children under age five to be about 9% and of enteropathogenic *E. coli* infections to be around 25% [34]. We explore opportunities for and factors that influence child exposure at multiple levels (e.g., community, household, individual), including multiple communities along an urban-rural gradient with a range of conditions to increase the applicability

to other LMICs. We also examine how household animal ownership influences exposure opportunities, which can provide important insights for potential mitigation strategies.

## Methods

### Study design and setting

We conducted qualitative research to understand if, why, how, and to what extent children are exposed to enteric pathogens in animal feces. To examine how community- and household-level factors may influence exposure, we interviewed mothers who owned and did not own animals. These mothers were participating in the Enteropatógenos, Crecimiento, Microbioma, y Diarrea (ECoMiD) study [34], a prospective cohort study in which mother-child dyads are followed from pregnancy through the critical first 24 months of life to examine how environmental exposures impact child gut microbiome composition and development. This study is reported in accordance with the Consolidated Criteria for Reporting Qualitative Research (COREQ) [35]. The location(s) where each of the 32 COREQ items is reported can be found in S2 Checklist, and our reflexivity statement can be found in S1 Text.

We carried out this work in seven ECoMiD study communities, representing four levels of varying urbanicity and rurality: (1) Esmeraldas (hereafter referred to as the urban community); (2) Borbón (a smaller town serving as a commercial center); (3) three rural villages near Borbón accessible by road (the rural road communities); and (4) two rural villages near Borbón only accessible by boat (the rural river communities) (Fig 1). The study area is primarily populated by Afro-Ecuadorians, with an increasing number of people of mixed race (mestizos) and a small number of Chachis, an indigenous group. Esmeraldas is an urban hub of the study area and capital of Esmeraldas Province, with a population of over 160,000 [36], It is densely populated and has the most access to WASH infrastructure, roads, and medical infrastructure.

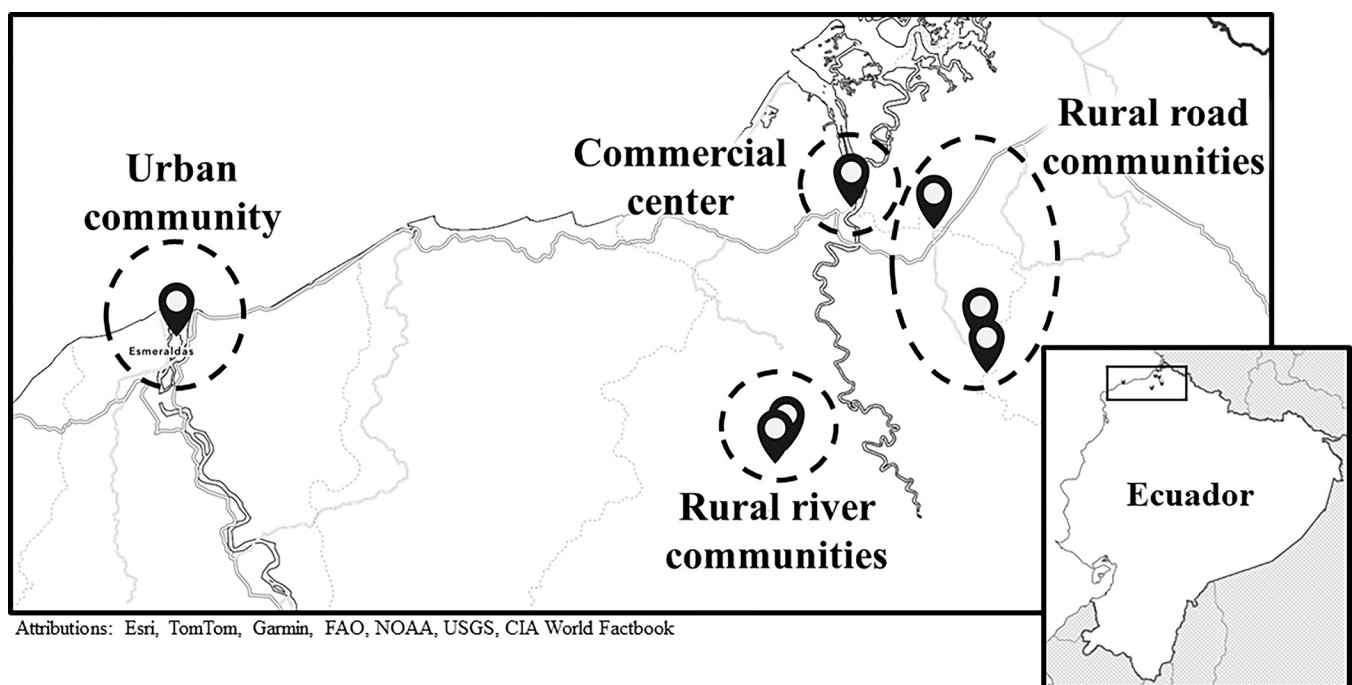

Attributions: Esri, TomTom, Garmin, FAO, NOAA, USGS, CIA World Factbook

**Fig 1. Map of the seven ECoMiD study communities where study participants were recruited.** The maps for this figure were created using ArcGIS Online [39], the Outline Map Basemap [40], and the World Countries Generalized layer [41].

Borbón is a town in Esmeraldas Province located at the confluence of the Cayapas, Santiago, and Onzole Rivers (population: 7,700) [36]. Borbón has underdeveloped infrastructure for its size, and basic WASH infrastructure of variable quality [37, 38]. We also conducted the study in three smaller communities with access to a road connecting them to Borbón and other trade hubs in the area, and two smaller, more remote with access to Borbón primarily via river. The rural road communities typically have more limited infrastructure, such as simple piped water systems, and the river communities predominantly rely on river water, wells, and rainwater [37, 38].

## Sample and participant selection

To examine how household-level factors may influence exposure, we enrolled two types of participants from the ECoMiD cohort: (1) mothers in households that owned at least one animal and (2) mothers in households that did not own animals. Our original study design called for 30 interviews with animal-owning mothers and 30 with non-animal-owning mothers, which was based on recommendations to conduct at least 16–24 in-depth interviews (IDIs) and to have a larger sample when studying complex topics [42, 43]. Mothers were eligible if their child in the cohort was between 6–18 months old. This age range was selected because children become more mobile and active during this time, making them particularly susceptible to environmental exposures. We used quota sampling to ensure an equal number of mothers who did and did not have animals in each of the four levels of urbanicity, if possible. To capture variability, we included mothers that owned different types and numbers of animals. Local study staff facilitated recruitment in each community by calling cohort mothers who had a child between 6–18 months old to query their animal ownership status and interest in participating.

## Data collection

Author BCA, who is a woman from Esmeraldas and has conducted qualitative research for more than 10 years, conducted go-along, semi-structured IDIs in Spanish from January 21st to April 21st, 2021. Go along IDIs enable simultaneous observation and interviewing as the interviewer and participant inhabit and engage with the spaces they are discussing [44, 45], which was ideal for our study objective. To understand how children may be exposed to enteric pathogens in animal feces, we asked mothers about a typical day for them and their child. Probes queried details about animals, environmental conditions, behaviors, and seasonality because interviews were conducted during the rainy season. The interview concluded with questions about reasons for and benefits of animal ownership and intra-household decision-making related to animals and the child. During interviews, mothers introduced the interviewer to household animals and showed where the animal(s) lived and spent time, as relevant. Basic demographics, household characteristics, and the type and number of animals (if any) owned by households were collected via a short survey. The IDI guide and short survey are provided in S1 Data. Qualitative Data Collection Tools. Systematic debriefing sessions were held between author AMB and BCA throughout data collection using a standard set of questions [46] to ascertain emerging themes in the data and enhance our approach in real time.

Interviews were audio recorded with permission from mothers. The go-along portions of interviews were not typically audio recorded due to logistical challenges (e.g., loud background noises, issues with audio recording due to social distancing requirements), but the interviewer took photos and detailed field notes about observations and the information ascertained during this portion of the interview. The audio recorded portion of interviews lasted 27 minutes on average (range: 15–50 minutes). Recordings were transcribed and de-identified by an

Ecuadorian, and then translated from Spanish to English verbatim by two other Ecuadorians. To standardize transcript formatting and obtain quality transcriptions and translations, we trained the translators on the research topic, interview content, conducting first-pass transcript reviewing while translating, and the goal of achieving meaning equivalence. Author AMB debriefed with translators after each of the initial five translations were completed and checked translations for accuracy. English translations were stored alongside the original Spanish transcripts, which allowed us to interact with the original and translated versions throughout analyses to conduct second-pass transcript reviews and to improve the rigor of our analyses. When mothers refused to be audio recorded (*n* = 17), the interviewer took detailed notes and created a transcript using the interview guide immediately following the interview. Mothers received an assortment of food items (e.g., rice, beans) as compensation for their time.

## Data analysis

To identify key themes in the data, we conducted thematic analysis using MaxQDA 2020 software (VERBI Software, Berlin, Germany). A codebook with deductive and inductive codes was developed iteratively throughout the analysis process using existing literature, transcript readings, and debriefing notes. To standardize our coding approach and ensure reliability, we double-coded two sets of five transcripts, cross-checking coding strategies and interpretation of data by each coder after each set. Subsequently, transcripts were double coded 10 at a time, after which coding agreement was checked to address inter-rater reliability issues. Then, the two coders systematically debriefed [46], resolved coding differences, and wrote memos on key themes. We did not calculate inter-rater agreement statistics to assess inter-rater reliability because coding was part of the process to discover themes, so agreement was not always the goal [47], and differences in coding style result in artificial low agreement [47, 48].

We assessed code and meaning saturation throughout the coding process [42, 43, 49] by tracking the number of additional codes and code definition changes there were after each round of coding (i.e., every 10 transcripts). Code saturation was considered achieved when 90% of meaningful codes were identified and developed, which occurred after coding five transcripts in this study. Meaning saturation was considered met when 90% of core codes had fully developed characteristics, which occurred after coding 10 transcripts. After coding, segments from transcripts for each code and intersections of prominent codes were queried and memos were written. Queries, memoing, and debriefing were performed iteratively to explore, describe, compare, conceptualize, and explain key themes–using the social ecological model [50] as a sensitizing construct to inform our interpretation and organization of the results throughout the analysis process [51]. Mothers' animal ownership status at the time of the interview was used to conduct comparative analyses. A description of major themes, along with their corresponding sub-themes and parent and child codes, is provided in S2 Data. Analytic Codes.

## Ethics

All participants provided written consent prior to data collection and received a copy of the consent form. Participants' rights to skip questions and end interviews at any time were emphasized by the interviewer. Institutional Review Boards at Emory University (IRB # 00101202) and Universidad San Francisco de Quito (IRB # 2018-022M and 021-011M) approved all study procedures. Additional information regarding the ethical, cultural, and scientific considerations specific to inclusivity in global research is including in the Supporting Information (S1 Checklist. Inclusivity Checklist).

## Results

Every child, regardless of household animal ownership, had opportunities to be exposed to enteric pathogens due to the ubiquity of animals and animal feces in the environment. Children had direct contact with animals and potentially came into contact with animal feces on surfaces, in environmental media, and on objects. Community norms and environmental factors and conditions influenced the quantity of animals and animal feces in the environment, as well as their proximity to children. Fig 2 summarizes these multi-level influences on potential child exposure that our data revealed, while Fig 3 provides a visual depiction of the influences across locations where children spend time daily.

The final sample consisted of 32 mothers in households with animals, and 28 without. Mothers were 28 years old on average (range: 19–47 years). Children were 10 to 18 months old and approximately half (52%, $n = 30$) were female. Our final sample did not include children between 6–10 months old because few children that age were enrolled in the cohort at the time of recruitment due to a pause in the study at the beginning of the COVID-19 pandemic. The type of water and sanitation access and animal ownership varied across communities. Sixty-six percent of households used water from an improved source for their child's drinking water and 81% had improved sanitation facilities. Over half of mothers who did not own animals (58%, $n = 15$) at the time of the interview had previously owned animals. Additional demographic information for the total sample and by study site are presented in Table 1.

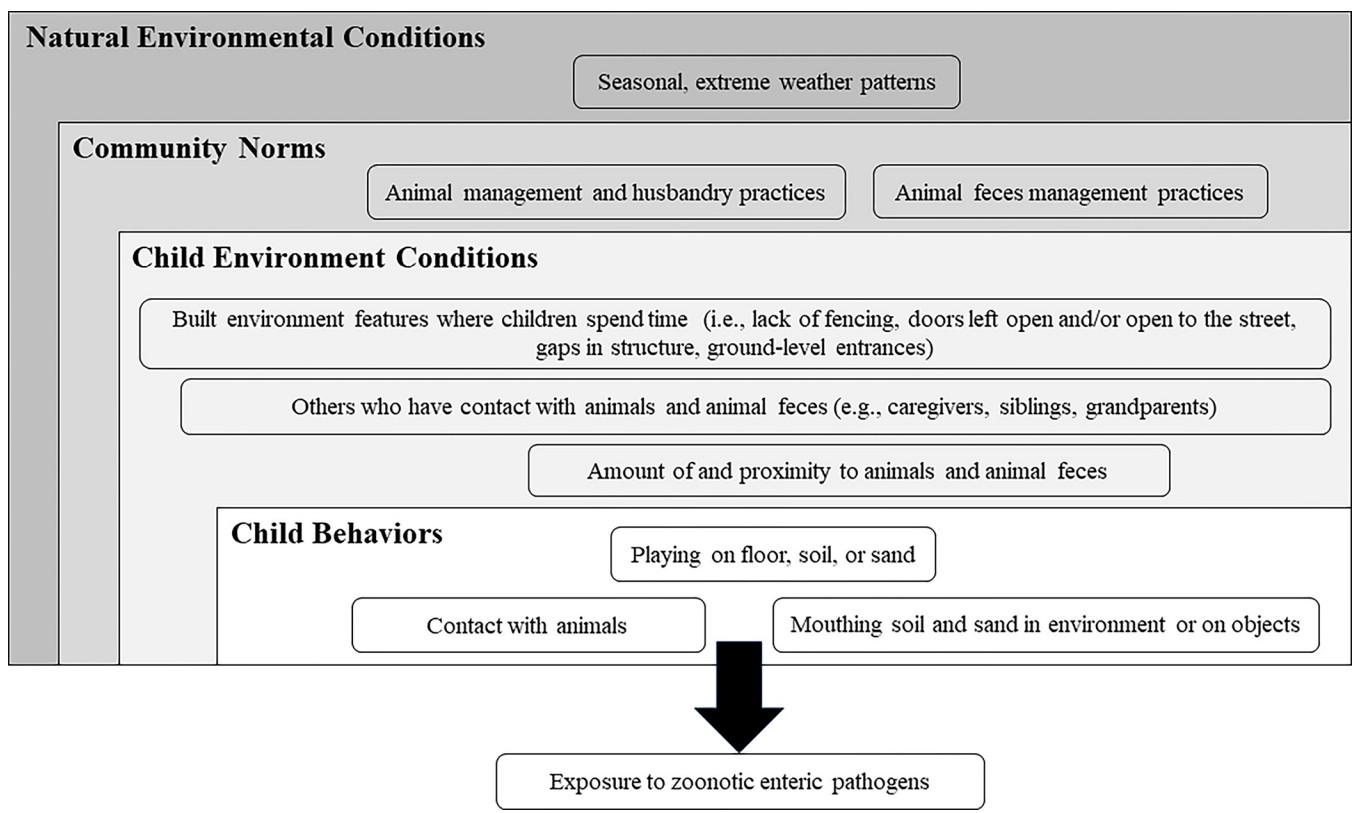

**Fig 2. Behaviors, conditions, and norms at multiple levels influence child exposure to enteric pathogens in animal feces.**

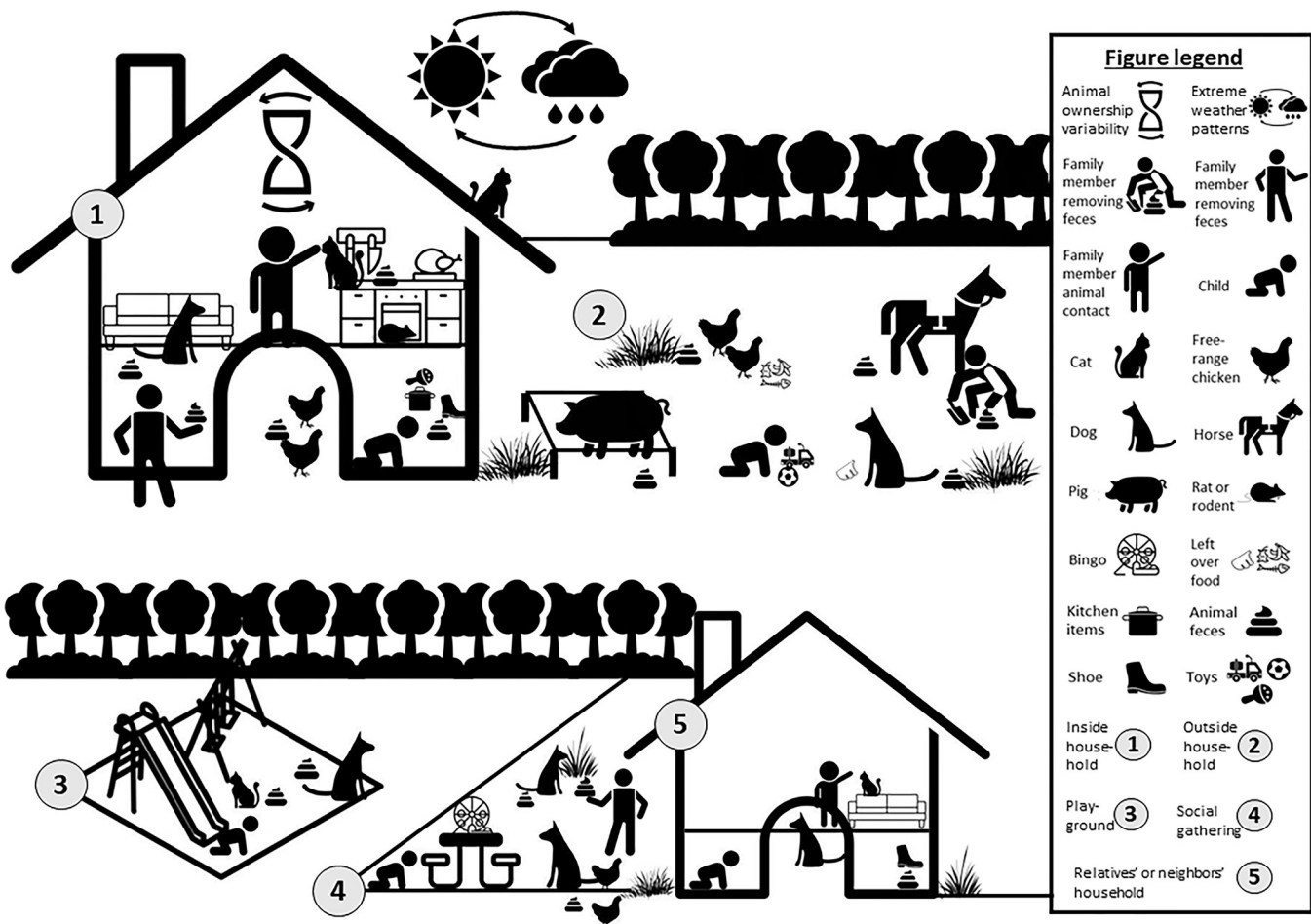

**Fig 3. Multi-level influences on potential child exposure to enteric pathogens in animal feces across locations where children spend time daily.**

## Child behaviors

Children performed behaviors daily that may lead to exposure to enteric pathogens inside and outside of their household and in other locations (e.g., parks, playgrounds, outdoor gatherings) where animals and animal feces were present. Children spent much of the day inside their house, though most played outside near their household regularly. They also habitually spent time in other locations, including at relatives' and neighbors' houses, community parks or playgrounds, and outdoor gatherings (e.g., bingo). Some mothers reported that child behaviors and interactions at home were quite different than those in other locations. For example, one child played in a garden by the river outside their grandmother's house, but largely played inside when they were at home.

> "It is different [at her grandmother's house] because she is not inside the house there. . .She goes to the river side and sits and observes, searches for stones, throws stones to the river, things like that."

-age 39, rural road community, non-animal owner

**Table 1. Mother-child characteristics and demographics for total sample and by the four study sites.**

| | Total | Urban | Commercial center | Rural road | Rural river |
|---|---|---|---|---|---|
| | 58 (100) | 18 (31) | 20 (34) | 14 (24) | 6 (10) |
| **Maternal characteristics** | | | | | |
| Age (years; mean [range]) | 28 (19–47) | 28 (19–47) | 26 (19–35) | 29 (21–39) | 30 (22–38) |
| Education (years; mean [range]) | 11 (0–17) | 11 (5–16) | 10 (0–16) | 12 (7–17) | 7 (0–12) |
| **Child characteristics** | | | | | |
| Age (months; mean [range]) | 14 (10–18) | 14 (10–18) | 14 (11–18) | 14 (10–17) | 16 (13–18) |
| Sex–male (n (%)) | 28 (48) | 11 (61) | 8 (40) | 4 (29) | 5 (83) |
| **Household characteristics** | | | | | |
| *Size* (mean [range]) | | | | | |
| # of total occupants | 5 (3–13) | 5 (3–8) | 5 (3–13) | 5 (3–10) | 5 (3–8) |
| # of children | 3 (1–6) | 3 (1–6) | 2 (1–4) | 3 (2–6) | 2 (1–4) |
| *Floor material* (n (%)) | | | | | |
| Cement | 32 (55) | 14 (78) | 7 (35) | 8 (57) | 3 (50) |
| Ceramic tile | 13 (22) | 3 (17) | 5 (25) | 4 (29) | 1 (17) |
| Wooden boards | 13 (22) | 1 (6) | 8 (40) | 2 (14) | 2 (33) |
| *Wall material* (n (%)) | | | | | |
| Cement or cement blocks | 52 (90) | 14 (78) | 14 (70) | 12 (86) | 4 (67) |
| Wooden boards | 10 (17) | 1 (6) | 6 (30) | 2 (14) | 2 (33) |
| Bricks | 3 (5) | 3 (17) | 0 (0) | 0 (0) | 0 (0) |
| *Roof material* (n (%)) | | | | | |
| Metal | 52 (90) | 15 (83) | 20 (100) | 11 (79) | 6 (100) |
| Cement | 3 (5) | 3 (17) | 0 (0) | 0 (0) | 0 (0) |
| Paving stone | 3 (5) | 0 (0) | 0 (0) | 3 (21) | 0 (0) |
| *Sanitation type* (n (%)) | | | | | |
| Indoor toilet connected to sewer systems, septic tank, or pit latrine | 47 (81)[a] | 14 (78) | 16 (80) | 13 (93) | 4 (67) |
| Indoor toilet that discharges to another location | 1 (2) | 0 (0) | 0 (0) | 0 (0) | 1 (17) |
| Pit latrine without a slab | 2 (3) | 0 (0) | 1 (5) | 0 (0) | 1 (17) |
| *Water source for child* (n (%)) | | | | | |
| Piped | 23 (40)[b] | 7 (39) | 10 (53) | 6 (43) | 0 (0) |
| Bottled/Purchased | 14 (24) | 1 (6) | 5 (26) | 8 (57) | 0 (0) |
| Tube well | 1 (2) | 0 (0) | 1 (5) | 0 (0) | 0 (0) |
| Public Tap | 11 (19) | 10 (56) | 1 (5) | 0 (0) | 0 (0) |
| River | 2 (3) | 0 (0) | 2 (10) | 0 (0) | 0 (0) |
| Rain | 6 (10) | 0 (0) | 0 (0) | 0 (0) | 6 (100) |
| *Owns animals* (n (%)) | | | | | |
| Any | 32 (55) | 11 (61) | 10 (50) | 7 (50) | 4 (67) |
| Dogs | 21 (36) | 9 (50) | 6 (30) | 4 (29) | 2 (33) |
| Cats | 21 (36) | 5 (28) | 6 (30) | 6 (43) | 4 (67) |
| Free-range chickens | 6 (10) | 1 (6) | 3 (15) | 2 (14) | 1 (17) |
| Production chickens | 3 (5) | 0 (0) | 3 (15) | 0 (0) | 1 (17) |
| Pigs | 4 (7) | 0 (0) | 2 (10) | 1 (7) | 1 (17) |

Data missing for 8 participants (4 urban, 3 semi-rural, 1, rural road)

Data missing for 1 semi-rural participant

Children had direct contact with animals across multiple locations, regardless of household animal ownership. Most interacted with dogs (e.g., petting, grabbing, and playing with them) that spent substantial amounts of time outdoors and were owned by their household, their relatives, or their neighbors. Some children had contact with cats (e.g., grabbing, touching, and carrying them), though most did not because mothers stated that cats "carry disease" and "cause asthma." A few mothers felt that contact with cats and dogs was beneficial for their child's immune system.

*"For my way of thinking, [animal contact] is so nothing will make her sick so that her body is adjusted to cats and dogs. . .so I tell her to touch them for her body's reaction. . ."*

-age 25, urban community, non-animal owner

Children played in indoor and outdoor spaces that were regularly contaminated with feces from dogs, cats, and free-range household chickens (hereafter referred to as free-range chickens), though no mother reported their child to have direct contact with animal feces. A few mothers who owned production chickens and pigs to generate household income also reported the presence of feces from these animals outside their household. Production chickens and pigs were kept outside near households contained within pens or pigsties, which also contained their feces. Children commonly crawled and walked freely, often unsupervised, throughout their house in the mornings and afternoons while mothers performed chores and cared for their other children. Free-range chickens and dogs were more active and reportedly entered households during mornings and afternoons, indicating that children may be in the same space as animals and their feces unsupervised. During this time, children played with toys and objects that they threw on the ground repeatedly, increasing the likelihood that objects and children's hands may become contaminated with animal feces and related enteric pathogens. For example, multiple mothers reported children playing on the bare floor with kitchen objects (e.g., pots, pans, spoons, glasses) that were later used for cooking and eating or to drink water. Other objects that children played with may have been contaminated with animal feces and related enteric pathogens because they were high-touch objects (e.g., television remotes, cell phones) or because of their functional purpose (e.g., a tool for cleaning).

*"The shoes [are her favorite toy]. And. . .what she likes to grab the most is also here in the kitchen. . .she grabs the pans or she starts to play with the trays. . .She grabs the broom, she puts it down and starts sweeping."*

-age 19, commercial center, non-animal owner

Children also played with toys, sticks, soil or mud, sand, rocks, and surface water outside near households and in public spaces. Some mothers reported toys becoming contaminated with animal feces when their child played with them outside.

## Child environment

The child environment refers to the close surroundings and daily conditions in which children lived, had direct contact with, and impacted their proximity and potential exposure to animal feces. This included aspects and conditions of the locations where children spent time and the individuals in close contact with them. Mothers reported that various individuals' contact with animals and animal feces, as well as features of the built environment where children spent time, influenced the proximity of children to animals and animal feces.

**Others' contact with animals and animal feces.** Mothers, siblings, extended family members, and other individuals who interacted with children had frequent contact with animals and/or animal feces, regardless of household animal ownership. Interactions with animals varied in intensity and included dogs, cats, free-range chickens, and pigs. Some mothers, extended family members, and other household visitors were reported to have intense contact with animals (e.g., raising chickens, bathing pigs).

*"I keep [the dogs and cats] clean so that they don't get fleas, ticks, or any of that. I wipe them down over there. They sleep in a dry place, and I keep the outside area where they poop clean. If I ever see a tick, they get an injection, or I wash them with chemicals."*

-age 28, urban community, owner of two dogs and two cats

However, most mothers had less intense interactions with animals. For example, some walked and played with family members' dogs and others fed leftover food to dogs, cats, and free-range chickens. Older siblings commonly had contact with and helped care for dogs and cats. Mothers found it more appropriate for their older children to interact with and care for animals.

Mothers', siblings', and extended family members' contact with animal feces across multiple locations was common. Mothers and grandparents had contact with feces from dogs, free-range chickens, and unspecified animals while removing it from where children play at their household and other locations (e.g., relatives' or neighbors' households, parks, playgrounds, outdoor social gatherings). Some mothers used a general "we" when reporting feces removal and disposal practices (e.g., "We throw it out."), suggesting that multiple household members have contact with animal feces and may contribute to contamination of children's interpersonal environment. Handwashing after removing feces was not always mentioned and surfaces were inconsistently cleaned with soap or disinfectants (e.g., bleach)–suggesting that environmental fecal contamination may remain after feces are removed. Household members also inadvertently stepped or put toys in animal feces, which resulted in feces contamination inside households.

*". . .Animal feces are brought inside, especially from dogs, on children's shoes. . .it can happen suddenly. There are remains [of animal feces outside] and children while playing at night do not see well and step in it and bring it in on their shoes."*

-age 26, rural road communities, owner of two dogs and one cat

Rat feces were observed in some kitchens by the interviewer, though mothers did not discuss this.

**Built environment features.** Household gaps, open doors, lack of fencing, and ground-level entrances influenced animals' ability to be present and defecate inside and outside near households. Mothers reported that cats–owned by the family, owned by others, and strays– were the most common animals in and near houses. Cats spent time on roofs and were difficult to keep out of houses because they could enter through any gaps or openings, especially at night. Some mothers made cats leave immediately while others gave them food even if they did not own them. Cats that were allowed to remain indoors spent time on living room furniture near where children played, under or on dining room tables where food was consumed, and/or in kitchens where food was prepared.

*"The cats come in and out of the house. They get under the bed, under the dining table. The dog also enters and leaves, but spends more time outside. . .The cats spend time in the kitchen,*

*on the floor, under the dog. . ."*

-age 32, rural road communities, owner of one dog and two cats

Mothers reported that they rarely found cat feces outside near their household, regardless of cat ownership, because it was buried in dirt or sand. However, cat-owning households reported finding feces inside on occasion.

Dogs were present frequently, but primarily outside near households, rather than indoors. Most dog-owning mothers reported that their dog(s) entered their house briefly on a typical day, to be fed by and spend time with their owners and/or because household doors were left open. During those times, dogs sometimes defecated inside. Mothers who did not own dogs rarely reported dogs entering their house and defecating. However, it was common practice for owners to let their dogs roam freely during the day, which allowed dogs to defecate outside near households that did not own dogs and lacked a fence. As a result, dog feces were found outside in household entryways or yards in the mornings and afternoons daily, regardless of ownership. Most mothers found multiple piles of dog feces near their household daily. In response, some tried to prevent dogs from being near their household. Others stated that they would put leftover food outside to feed the dogs and avoid food waste, which encouraged the presence of dogs regardless of ownership status.

*"I don't allow [animals to get near the house] because they get used to it. There is a dog that knew that I put food out sometimes so as not to throw it away, but I don't give it to him anymore because they get used to it."*

-age 25, rural road communities, non-animal owner

Free-range chickens and their feces were found outside near many households throughout the morning. The lack of fencing around households allowed free-range chickens to roam from compound to compound and defecate near households that did and did not own animals. When household doors were left open, nearby chickens reportedly entered and defecated inside some households. A few mothers reported that the raised entrance to their house prevented free-range chickens from entering. For example, one mother's house was raised up on stilts, making the house's door approximately three meters off the ground. Most mothers did not actively deter the presence of free-range chickens, and some encouraged their presence by putting leftover food outside, similar to dogs.

At other locations where children spent time, the same features of the built environment (i.e., household gaps, open doors, lack of fencing, and ground-level entrances) allowed cats, dogs, and free-range chickens and their feces to be proximal to children. In some cases, the types of animals and animal feces that were present in other locations differed from the child's household. For example, some relatives owned animals that were not present at the child's household.

*". . .At my mom's house, she has like nine dogs and like two cats. So [my child] is over there, and my sister brings her up so she spends time with the dogs and playing with my nephews that are also there."*

-age 22, urban community, owner of one cat

Similar to trends in children's homes, animals were often present and defecated near where children spent time, such as in relatives' households and in parks. Most mothers observed dog feces outside others' households, in the street where children played, and at parks. Some reported the presence of cat and free-range chicken feces in these other locations, though this

was less common. Cats, dogs, and free-range chickens also entered inside relatives' and neighbors' households due to structural gaps, lack of fencing, and open doors. However, indoor fecal contamination in others' households was similar to indoor contamination at children's homes: infrequent and largely from shoes or objects.

> *"We always clean at my mother's place because the children play outside a lot. As I say, I go to my mother's house every day. Sometimes we sweep together... [the animal feces] are picked up, but I would say yes [feces are brought inside] because sometimes they step on it without noticing."*

-age 27, rural road community, non-animal owner

## Community norms

Community norms refer to sets of behaviors or practices that are widely accepted and expected within a community, including those related to collective responsibility, health and well-being, and environmental hygiene, among others. Norms that influenced child proximity to animals and animal feces pertained to the animal management and husbandry practices and feces management practices at a child's household and throughout their community.

**Animal management and husbandry.** It was common for animal owners in the study area to let their cats, dogs, and free-range chickens roam freely throughout communities. As described above, this practice allowed animals to move from compound to compound and defecate in or near many places where children spent time.

> *"The dog, cat, and chickens all spend time outside the house...The cat is in the brush. The chickens, like, they are being raised freely. They just walk through the town. They come back at night to sleep, and the cat does too. [The cat] practically just comes in at night."*

-age 31, commercial center, owner of 15 free-range chickens, one dog, and one cat

Free-range practices were perceived to be healthier for the animals and helped offset the economic burden of feed by allowing animals to forage for food. Mothers reported that people who owned dogs let them roam free during the day specifically to urinate, defecate, and/or forage for food. Free-range chickens were also released during the day to forage for food, and then placed in enclosures inside or outside near households in the afternoon or at night for protection against predators and theft. Uniquely, cats were largely active at night and would leave their owners' house in search of food. In contrast to cats, dogs, and free-range chickens, pigs and production chickens used to generate household income were contained within pens or pigsties, which prevented them from roaming throughout communities.

> *"[The pigs are raised] in the pigsty...because if we let them loose, they walk around in other people's patios daringly and there are people that don't like that. Or the pigs can get sick, so that's why they live locked up there."*

-age 35, rural river community, owner of four free-range chickens, four dogs, two pigs, and four cats

**Animal feces management.** Mothers used multiple and varying animal feces management practices that contributed to the contamination of their household and the surrounding community environment. Practices depended on the type of animal feces, the animal's

defecation location, and other factors (e.g., availability of trash collection services), which are described in Table 2. Management differences by animal type were related to the frequency of defecation, the different types and sizes of stool, and whether the animal buried their feces. For example, mothers rarely managed cat feces because cats buried their feces, whereas dog feces were abundant and a noticeable nuisance that mothers removed from their yard regularly. Mothers removed and disposed of dog feces in the trash when garbage collection services were available, and reported rinsing or throwing feces into the surrounding area when garbage collection was irregular, infrequent, or unavailable or if feces were dried out.

> *"The poop is collected with a shovel and thrown directly into the surrounding vegetation because sometimes [the dog feces] are already dry and I do not wait for the garbage cart to throw them away…"*

-age 31, commercial center, non-animal owner

Feces from free-range chickens, production chickens, and pigs were also regularly rinsed away using a hose or bucket of water or thrown into the surrounding area, potentially spreading fecal contamination rather than eliminating it.

**Table 2. Community-level animal feces removal and disposal practices and factors that influence their use.**

| Type of feces | Removal and/or disposal practices | Factors influencing practices | Example |
|---|---|---|---|
| **Cat** | • Collected via trash service<br>• Threw into nearby vacant lot<br>• Threw into surrounding vegetation<br>• Buried with soil or sand | • Feces dried out when found<br>• When/if trash pickup would occur<br>• Location found (e.g., inside home, outside, etc.)<br>• Proximity to vacant lot or river<br>• Pandemic conditions/awareness | "We throw the cat feces away to the trash…but now let's say, we are constantly cleaning. We throw chlorine because of what we are going through [with the pandemic]."<br>• age 35, semi-rural community, owns one dog and one cat |
| **Dog** | • Collected via trash service<br>• Rinsed away with water<br>• Threw into septic tank<br>• Threw into nearby vacant lot<br>• Threw into surrounding vegetation or area, including rivers<br>• Buried with soil or sand | • Feces dried out when found<br>• When/if trash pickup would occur<br>• Location found<br>• Proximity to vacant lot or river<br>• Other animal owners' practices | "We throw it out because the owner doesn't pick it up… [we throw it] out in front where there's that piece of land. That's where we throw it."<br>• age 32, urban community, non-animal owner |
| **Free-range chicken** | • Threw into nearby vacant lot<br>• Threw into surrounding vegetation, including rivers<br>• Buried with soil or sand<br>• Rinsed away with water, including letting rain wash it away<br>• Collected and stored for fertilizer | • Feces dried out when found<br>• When/if trash pickup would occur<br>• Location found<br>• Proximity to vacant lot or river<br>• Other animal owners' practices<br>• Feces contained to cage or spread throughout environment<br>• Had a use for fertilizer | "There is chicken feces in the yard. The yard is open and the neighbor has some chickens and they go in the yard. I don't know how much feces because [the neighbor] knows how to clean. She scoops it up or covers it with dirt and I don't always realize it."<br>• age 25, rural road communities, non-animal owner |
| **Production chicken** | • Collected via trash service<br>• Threw into surrounding vegetation | • When/if trash pickup would occur<br>• Feces contained to cage or spread throughout environment | "…The feces dry out and mix with the sawdust and it is not eliminated daily. The feces are thrown away with the sawdust and we change it one or two times a week."<br>• age 31, semi-rural community, owns three production chickens |
| **Pig** | • Rinsed away with water | • Feces contained to pigsty<br>• Proximity to river | "We throw [the pig feces] away by the 'plan,' a ravine."<br>• age 35, rural river communities, owns four free-range chickens, four dogs, four cats, and two pigs |
| **Horse** | • None | • Other animal owners' practices | "From horses, it occurs usually two or three times daily. When they poop, the owner comes down and cleans it. When they are in a hurry, they leave it."<br>• age 36, rural river communities, owns one cat |

Regardless of animal type, feces on floors inside the home or in soil outside near child domestic and play areas were removed quickly most of the time, in hopes of allowing children to play in feces-free environments. Removing feces from outdoor household and public play spaces was less common and more difficult due to free-range animals, resulting in child toys and shoes becoming contaminated with feces.

## Natural environmental conditions

Natural environmental conditions refer to the atmospheric characteristics of a geographic area based on its climate and seasonal weather patterns, and include temperature and precipitation, among others. Extreme weather events, such as heavy rain and flooding that commonly occurred during the rainy season, influenced the number of animals present and their proximity to children. Some mothers reported that animals died from drowning during this time. Others reported bringing their dogs inside their house more often to escape bad weather. Additionally, the number of free-range chickens in communities and near children varied by season because of flooding.

> *"[During the winter], it rains too much. And when the river grows, it floods all the town and the houses are practically sunk. . .So we can't raise [chickens] like that because they have nowhere to run. . .During the summer, we can raise them better, but now in the winter, it is not possible."*

> -age 25, rural river communities, non-animal owner

## Discussion

By qualitatively characterizing the interrelated community, household, and child factors that drive exposure to enteric pathogens in animal feces, we identified critical insights for the development of effective mitigation strategies. We found that animals and animal feces were ubiquitous–regardless of animal ownership–due to community- and household-level animal and feces management practices. Although 66% of households had access to improved drinking water sources and 81% had improved sanitation facilities, all mothers reported opportunities for their child to be exposed to animal feces, even though 45% of households did not own animals. These findings are in line with other studies showing that WASH interventions aiming to reduce enteric infections by targeting human feces alone likely overlook other significant sources of environmental fecal contamination and enteric pathogens [19, 22, 24, 52–54]. The findings also highlight that focusing only on animal ownership and only on household environments provide insufficient information to identify the relative risk of child exposure to enteric pathogens in animal feces. Collectively, our results suggest that reducing enteric pathogen transmission will require integrated programming that targets both human and animal feces and addresses the multilevel, upstream drivers of environmental fecal contamination and child exposure. Below we highlight two key findings and their implications.

First, free-range animal management and husbandry practices at the community- and household-level resulted in the persistent presence of animals and fecal contamination in children's environments, including at their household and at other locations (e.g., parks, playgrounds). Existing evidence clearly demonstrates that child household and play spaces can be contaminated with animal feces and associated enteric pathogens when free-range animals are present [27, 32, 55–57], and that proximity to animals and such contaminants increases the risk of enteric infections among children [29, 31, 52–54, 58–60]. Our findings add to existing research by highlighting that child exposure to enteric pathogens in animal feces is not solely

shaped by household practices but is also significantly shaped by the normative practices of others in their community. Findings further suggest that sole focus on the household environment provides inaccurate and/or incomplete data because other significant locations where children may be exposed to enteric pathogens in animal feces can be missed. Consequently, research and interventions exclusively targeting the household level may be insufficient to examine and reduce child exposure to animal feces.

Second, inadequate animal feces management (AFM) practices, which include behaviors beyond removal and disposal, contributed to fecal contamination of children's environments. While animal feces near child domestic and play areas were removed the majority of the time, maternal handwashing was seldom discussed and surfaces were inconsistently cleaned with soap or disinfectants after removal of animal feces. In Bangladesh, maternal hand contamination with animal feces was near universal despite the reported use of tools to clean up animal feces [61] and in rural India, removing child feces without using tools was associated with increased hand contamination [62, 63]. Findings from these studies suggest that handwashing after handling animal feces could help reduce child exposure to enteric pathogens, especially given studies showing strong correlations between animal feces and enteric pathogen contamination on caregiver and child hands in the same household [61, 64]. Similarly, cleaning or disinfecting surfaces after removal could reduce exposure, as a study in rural India found that environmental fecal contamination remains even after child feces are removed [62]. The modalities by and locations where mothers reported disposing of animal feces in our study–such as rinsing feces into drains or ditches and throwing feces into surrounding vegetation–also have been shown to increase environmental contamination in studies on child feces. [62, 63, 65] and can intensify transmission through various pathways. As a result, inadequate AFM practices in one household could impact the environmental contamination and exposure of children in neighboring households.

Taken together, these findings suggest that integration of safe AFM practices with existing child feces management programming is an important area for future research. Research could enable identification of integrated exposure control approaches that capture the many enteric pathogen exposure pathways related to both animal and human feces. Research on child feces management suggests that unsafe practices along the feces management pathway–which includes defecation, feces removal and disposal, defecation location cleaning, anal cleansing, and handwashing–increase environmental contamination [62, 63, 65, 66]. The AFM practices identified in this study are similar to child feces management practices reported elsewhere [62, 63, 65, 66], and previously observed among mothers in the ECoMiD cohort [66]. Incorporation of safe practices along the AFM pathway, similar to those developed for child feces management, (e.g., remove feces using a tool, clean defecation location with soap and water) [65], may therefore be an effective, practical approach for intervening on the multiplicity of exposures related to various animal sources. To establish safe AFM practices, future research should assess unsafe practices and feces contamination along the pathway established for child feces using surveys, observation or spot checks, and environmental sampling. The established child feces management pathway can guide data collection at key points to validate the AFM pathway, with changes made as relevant.

These results demonstrate that multilevel, multisectoral interventions to mitigate child exposure to enteric pathogens in animal feces are needed. Existing interventions have overwhelmingly included a single-component and primarily focused on the individual child- or household-level (e.g., building and encouraging use of enclosures to contain animals and animal feces, providing and encouraging use of child playpens to minimize their contact with environments contaminated by animals, etc.) [18, 67]. However, intervening on exposure to enteric pathogens in animal feces–like human-sourced enteric pathogen exposures–requires

the disruption of the upstream causes of environmental fecal contamination and the multiple exposure pathways to diverse enteric pathogens. Approaches that target exposure factors at multiple levels should consider features of the natural and built environment, as well as aspects of community, household, and child practices and behaviors. Such approaches could be integrated into existing comprehensive WASH interventions that aim to mitigate exposure to human feces.

## Strengths and limitations

This study used rigorous qualitative methods (i.e., analyzing verbatim transcripts, double coding, systematic debriefing, and assessment and achievement of meaning and code saturation) that strengthen the validity of findings [42, 43, 68, 69]. It included participants from multiple communities along an urban-rural gradient with a range of conditions, increasing the generalizability of our findings. The sample sizes were however uneven across communities due to circumstances surrounding the COVID-19 pandemic. Still, we found that the overall sample size was sufficient for saturation [43]. Additionally, our final sample only included children between 10–18 months old, which could limit the applicability of our results to children aged 6–10 months. Lastly, reliance on mothers may have biased our findings because they were not always the main or sole caregiver on a typical day and could have provided incomplete or inaccurate information about their child. However, the use of go-along IDIs enabled simultaneous in-depth interviewing and observation of the child's environment, which ascertained key details that were not reliant on maternal reporting. Reflections, observations, and analyses conducted by author BCA, an experienced local researcher, strengthened interpretation by providing insights that enhanced the credibility of community comparisons and information provided by mothers [42, 68].

## Conclusions

Current approaches to control enteric pathogens that center on individual- or household-level interventions are likely insufficient to address the multifaceted nature of exposure to enteric pathogens in animal feces. Future mitigation strategies should adopt a broader approach that considers the multilevel nature of exposure, including factors at the community, household, and child levels. Such strategies will need to distinguish between and address both the upstream causes of environmental contamination and more proximal causes of enteric pathogen exposure in their design to enable interventions to be more targeted and effective.

## Supporting information

**S1 Checklist. Inclusivity checklist.**
(DOCX)

**S2 Checklist. COREQ checklist.**
(DOCX)

**S1 Text. Reflexivity statement.**
(DOCX)

**S1 Data. Qualitative data collection tools.**
(DOCX)

**S2 Data. Analytic codes.**
(DOCX)

## Acknowledgments

We would like to thank local field staff for their help with recruiting, logistics, and data collection for this project. We would also like to thank all the participants in Ecuador that made this project possible.

## Author Contributions

**Conceptualization:** April M. Ballard, Regine Haardörfer, Matthew C. Freeman, James Trostle, Joseph N. S. Eisenberg, Gwenyth O. Lee, Karen Levy, Bethany A. Caruso.

**Data curation:** April M. Ballard, Betty Corozo Angulo.

**Formal analysis:** April M. Ballard, Betty Corozo Angulo, Nicholas Laramee, Jayden Pace Gallagher.

**Funding acquisition:** Joseph N. S. Eisenberg, Gwenyth O. Lee, Karen Levy, Bethany A. Caruso.

**Investigation:** April M. Ballard, Betty Corozo Angulo.

**Methodology:** April M. Ballard, Betty Corozo Angulo, Regine Haardörfer, Bethany A. Caruso.

**Project administration:** April M. Ballard, Betty Corozo Angulo.

**Resources:** April M. Ballard.

**Software:** April M. Ballard.

**Supervision:** April M. Ballard, Regine Haardörfer, Matthew C. Freeman, James Trostle, Joseph N. S. Eisenberg, Gwenyth O. Lee, Karen Levy, Bethany A. Caruso.

**Validation:** Betty Corozo Angulo, Nicholas Laramee, Jayden Pace Gallagher.

**Visualization:** April M. Ballard.

**Writing – original draft:** April M. Ballard.

**Writing – review & editing:** Betty Corozo Angulo, Nicholas Laramee, Jayden Pace Gallagher, Regine Haardörfer, Matthew C. Freeman, James Trostle, Joseph N. S. Eisenberg, Gwenyth O. Lee, Karen Levy, Bethany A. Caruso.

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
