## [Decision Letter · Decision Letter 0]

15 Apr 2024

PGPH-D-24-00231

Multilevel factors drive child exposure to enteric pathogens in animal feces: A qualitative study in northwestern coastal Ecuador

Dear Dr. Karen Levy,

Thank you for submitting your manuscript to PLOS Global Public Health. After careful consideration, we feel that it has merit but does not fully meet PLOS Global Public Health’s publication criteria as it currently stands. Therefore, we invite you to submit a revised version of the manuscript that addresses the points raised during the review process.

We look forward to receiving your revised manuscript.

Kind regards,

Muhammad Asaduzzaman, MD MPH MPhil

Academic Editor

Journal Requirements:

1. Please include a complete copy of PLOS’ questionnaire on inclusivity in global research in your revised manuscript. Our policy for research in this area aims to improve transparency in the reporting of research performed outside of researchers’ own country or community. The policy applies to researchers who have travelled to a different country to conduct research, research with Indigenous populations or their lands, and research on cultural artefacts. The questionnaire can also be requested at the journal’s discretion for any other submissions, even if these conditions are not met.  Please find more information on the policy and a link to download a blank copy of the questionnaire here: https://journals.plos.org/globalpublichealth/s/best-practices-in-research-reporting. Please upload a completed version of your questionnaire as Supporting Information when you resubmit your manuscript.”

b. If any authors received a salary from any of your funders, please state which authors and which funders.

If you did not receive any funding for this study, please simply state: “The authors received no specific funding for this work.

Additional Editor Comments (if provided):

Reviewers' comments:

Reviewer's Responses to Questions

**Comments to the Author**

1. Does this manuscript meet PLOS Global Public Health’s publication criteria? Is the manuscript technically sound, and do the data support the conclusions? The manuscript must describe methodologically and ethically rigorous research with conclusions that are appropriately drawn based on the data presented.

Reviewer #2: Yes

Reviewer #3: Yes

Reviewer #4: Yes

Reviewer #5: Yes

2. Has the statistical analysis been performed appropriately and rigorously?

Reviewer #2: N/A

Reviewer #3: Yes

Reviewer #4: N/A

Reviewer #5: N/A

3. Have the authors made all data underlying the findings in their manuscript fully available (please refer to the Data Availability Statement at the start of the manuscript PDF file)?

Reviewer #2: Yes

Reviewer #3: Yes

Reviewer #4: Yes

Reviewer #5: Yes

4. Is the manuscript presented in an intelligible fashion and written in standard English?

Reviewer #2: Yes

Reviewer #3: Yes

Reviewer #4: Yes

Reviewer #5: Yes

5. Review Comments to the Author

**Reviewer #2: SUMMARY OF THE RESEARCH AND REVIEWERS OVERALL IMPRESSION**

This article authored by Ballard et al., discusses an interesting topic on the risks of children aged 6–18 months being exposed to animal faeces, which can lead to exposure to harmful enteric pathogens. It emphasises the importance of combining animal and child faeces management programmes to decrease the risk of exposure to these pathogens, which are linked to various health issues commonly seen in childhood. The article is impressive for its rigorous methods and its substantive contribution to the existing body of knowledge in this field. However, there are some specific areas for improvement.

SPECIFIC AREAS FOR IMPROVEMENT

1. Abstract

Lines 13–14: The abstract should emphasise the observations made in the study rather than compare them to previous findings. The discussion section would be a more appropriate place for that.

2. Introduction

The introduction is perfectly put- together. However, the authors should consider including statements about the prevalence of infectious and diarrheal diseases in the region where the study took place.

Line 37: The authors should specify which enteric pathogens are being considered here.

3. Sample And Participant Selection

This section should be edited to focus on the sample selection and the criteria used for inclusion and exclusion. Additionally, the results presented in Table 1 should be relocated to the results section.

The abstract indicates that the study included children aged 6–18 months, yet the methods section highlights that children aged 6–10 months were excluded. May the authors clarify to ensure that the age ranges are clearly defined?. The final sample only included those aged 10–18 months; therefore, the study is considered to be focused on that age group.

4. Data Collection

Line 126: There is a need to provide sufficient detail on how the translation process was conducted to ensure reliability.

5. Strengths and Limitations

The authors should restructure the section to present the strengths first, followed by the limitations. In its current state, the section is somewhat cluttered. Sample selection limitations should be emphasized, as the exclusion of those aged 6 - 10 months surely would have an impact on the interpretation of the study results.

**Reviewer #3: **This article brings a new insight in child exposure to enteric pathogens through animals' feces. This should be takeb into account in the effort for child protection against oathogen. As this is a qualitative study, my main recommendation to the author is to add more verbatims from mothers' opinion to support the summaries written under different level of exposureaspects of the study. Below are my specific comments on this manuscript:

Comment #1

Line 103. Table 1.

Under child characteristics

Sex – male : It is supposed that figures pertaining to Sex-male are absolute frequency. The authors need to add precision in the first column about figures between brackets (i.e. (48) ?)

Comment #2 [line 103 Table 1]

To make the table 1 clearer and more readable, there a need to insert a title line Under Household characteristics to introduce the size

Size (mean [range]) as it is presented for other characteristics. This will avoid mention of (mean [range]) for each line.

Size (mean [range])

Household size 5 (3-13) …

# of children 3 (1-6) …

Comment #3 [line 103 Table 1]

“Owns animals (n(%))“ can be replaced by “Pet ownership (n(%))”

Figures on the same line with the variable name should be written below the variable name by inserting a line with a category that can be stated as

“All/overall” 32 (55) 11 (61) 10 (50) 7 (50) 4 (67)

There a need to add more verbatims to make mother talk about what is summarized by the authors:

• At the end of the section entitled “Child environment”, it can be inserted starting on line 217

• In “Others’ contact with animals and animal feces”, a verbatim cab be added at line 222 after the full stop. The sentence starting with “However,” can be moved after this verbatim.

• The sentence on line 267 to 269 needs to be supported by verbatims from mothers’ opinion on this statement.

• The sentence on lines 289 and 290 needs to be supported by verbatim from mothers. It can be inserted below line 293.

• Below line 299, there is a need to illustrate how mothers mentioned community norms and how do community norms influence child exposure to animals’ feces.

• One or two verbatims should be added to support authors last summary on “animal management and husbandry”. They can be inserted below the line 334.

**Reviewer #4:** I would like to thank the authors for their manuscript. Understanding the factors of child exposure to enteric pathogens is a critical topic to help reduce the disease burden of enteric pathogens. This study which focuses specifically on communities in Ecuador can provide important lessons for countries with similar settings. The authors have written a good Introduction which helps set the stage for the study. However, the methodology requires additional information to enhance transparency and reproducibility, and the authors need to re-assess how to present the results section so it answers the objectives of the study.

Although this is a good first draft, I have additional suggestions, comments, and edits to enhance the manuscript. Please refer to my comments below for a full review.

Introduction:

Line 23: “are the fifth leading cause of death in children under age five”. Please clarify whether this is globally, nationally or regionally.

Lines 51-57: The authors should consider providing 1-2 lines on their rationale for conducting the study in Ecuador (example - burden of enteric infections etc.)

Methodology:

I would recommend authors follow Consolidated Criteria for Reporting Qualitative Research (COREQ) checklist for reporting qualitative research. While the authors have provided details on their methodological approach, some additional details are required. The authors should explicitly mention how the study adheres to the COREQ guidelines for reporting qualitative research. COREQ includes specific reporting criteria related to research team & reflexivity, participant selection, data collection, and data analysis, which would enhance the transparency and quality of the study. Please follow the sub-headings reported in COREQ for updating the methods section.

Line 63: Would recommend fully defining ECoMiD (Enteropatogenos, Crecimiento, Microbioma y Diarrea.) Please also mention that is a prospective cohort study.

Line 66-70: It would be great if the authors could include a geographic map of the seven study locations.

Table 1: Row Education – Please clarify what the unit of measure is? Years? Grade?

Line 107-108: Under research team and reflexivity, what was the positionality of BCA and what impacts might this have had on the findings? A lot of this information can be detailed in Supplementary Methods.

Lines 108-110- Please provide the semi-structured Interview Guide in the Supplementary Methods for transparency and reproducibility.

Lines 116-118: I’m assuming all the information on basic demographics, household characteristics etc. is what is shown in Table 1. Would be helpful for authors to provide the survey questionnaire in the Supplementary File as well.

Lines 118-120: Would recommend moving this to Data Analysis section.

Lines 121-128: Please indicate how the transcripts were generated and how accurate was this transcription process? Was this done using software or manually? How were transcripts checked for validity, reliability, and completeness?

Lines 132-134: Using a Table, please provide the themes/sub-themes and their description that emerged from the thematic content analysis. Please include this codebook in Supplementary Files for transparency and reproducibility.

Results

In the introduction, the authors define their objectives as “explore the interrelated community, household, and individual child factors that influence exposure. We explore opportunities for and factors that influence exposure across multiple communities along an urban-rural gradient with a range of conditions to increase the applicability to other LMICs. We also examine how household animal ownership influences exposure opportunities, which can provide important insights for potential mitigation strategies.”

I would strongly recommend that authors re-arrange their results section to follow their objectives in sequential order for readability.

For example, I don’t see clear results on the opportunities for and factors that influence exposure by rural and urban settings, as the authors have mentioned in their introductory objectives. Are there any stark differences seen in results in Table 2. Community-level animal feces removal and disposal practices and factors that influence their use, when stratified by urban and rural settings?

While the results are very interesting and rich, it’s hard to tease out results for each of the objectives in the study. I would suggest that the authors re-consider and re-arrange the results on how to explain their findings, so it tells a cohesive story. In its current form, the results feel disjointed and not linked to the three study objectives.

The resolution on both figures needs to be higher.

Discussion:

I would recommend authors discussing the findings of related to other papers in the literature, including quantitative studies.

1. Baker KK, Mumma JAO, Simiyu S, Sewell D, Tsai K, Anderson JD, MacDougall A, Dreibelbis R, Cumming O. Environmental and behavioural exposure pathways associated with diarrhoea and enteric pathogen detection in 5-month-old, periurban Kenyan infants: a cross-sectional study. BMJ Open. 2022 Oct 31;12(10):e059878.

2. Knee J, Sumner T, Adriano Z, Berendes D, de Bruijn E, Schmidt WP, Nalá R, Cumming O, Brown J. Risk factors for childhood enteric infection in urban Maputo, Mozambique: A cross-sectional study. PLoS Negl Trop Dis. 2018 Nov 12;12(11):e0006956.

3. Chard AN, Levy K, Baker KK, Tsai K, Chang HH, Thongpaseuth V, Sistrunk JR, Freeman MC. Environmental and spatial determinants of enteric pathogen infection in rural Lao People's Democratic Republic: A cross-sectional study. PLoS Negl Trop Dis. 2020 Apr 8;14(4):e0008180.

Another limitation of the study could be the bias introduced by interviewers during the key informant interviews due to probing or leading questions. Please address this in the manuscript as well.

**Reviewer #5:** General comment: The authors have presented an interesting, well-written manuscript that emphasizes an important topic in child health with a good flow.

Review comments:

1. The manuscript should include country-specific information on diarrhea caused by enteric pathogens in children under five. Including this information will inform the readers of the prevalence of the problem in Ecuador. It's important to include data from both human and animal health sources to provide a comprehensive understanding of the issue.

2. The first question that came to mind while reading was why the age restriction was 6-18 months. It is noted that this study was part of the ECoMid study, which had children within that age range. However, unless the reader goes to the reference for the ECoMid study protocol, it would be useful for the author to expand a bit on this selection of age and why not 5 years and under.

3. Was there a question that asked to the mothers or guardians whether the child had experienced episodes of diarrhea in the recent past?

4. Table 1: Maternal characteristics, Education (mean [range]): It is not clear what those figures represent.

5. Attaching the questionnaires (English version) as supplementary information would be helpful to understand how the data was collected.

6. PLOS authors have the option to publish the peer review history of their article (what does this mean?). If published, this will include your full peer review and any attached files.

**Do you want your identity to be public for this peer review?** For information about this choice, including consent withdrawal, please see our Privacy Policy.

Reviewer #2: No

Reviewer #3: No

Reviewer #4: No

Reviewer #5: No

---

## [Decision Letter · Decision Letter 1]

17 Jul 2024

PGPH-D-24-00231R1

Multilevel factors drive child exposure to enteric pathogens in animal feces: A qualitative study in northwestern coastal Ecuador

Dear Dr. Karen Levy,

Thank you for submitting your manuscript to PLOS Global Public Health. After careful consideration, we feel that it has merit but does not fully meet PLOS Global Public Health’s publication criteria as it currently stands. Therefore, we invite you to submit a revised version of the manuscript that addresses the points raised during the review process.

We look forward to receiving your revised manuscript.

Kind regards,

Muhammad Asaduzzaman, MD MPH MPhil

Academic Editor

Journal Requirements:

Additional Editor Comments (if provided):

Reviewers' comments:

Reviewer's Responses to Questions

**Comments to the Author**

1. If the authors have adequately addressed your comments raised in a previous round of review and you feel that this manuscript is now acceptable for publication, you may indicate that here to bypass the “Comments to the Author” section, enter your conflict of interest statement in the “Confidential to Editor” section, and submit your "Accept" recommendation.

Reviewer #3: All comments have been addressed

Reviewer #4: (No Response)

Reviewer #5: (No Response)

2. Does this manuscript meet PLOS Global Public Health’s publication criteria? Is the manuscript technically sound, and do the data support the conclusions? The manuscript must describe methodologically and ethically rigorous research with conclusions that are appropriately drawn based on the data presented.

Reviewer #3: Yes

Reviewer #4: Yes

Reviewer #5: Yes

3. Has the statistical analysis been performed appropriately and rigorously?

Reviewer #3: Yes

Reviewer #4: Yes

Reviewer #5: Yes

4. Have the authors made all data underlying the findings in their manuscript fully available (please refer to the Data Availability Statement at the start of the manuscript PDF file)?

Reviewer #3: Yes

Reviewer #4: Yes

Reviewer #5: Yes

5. Is the manuscript presented in an intelligible fashion and written in standard English?

Reviewer #3: Yes

Reviewer #4: Yes

Reviewer #5: Yes

6. Review Comments to the Author

Reviewer #3: I am fully satisfied with this corrected version of this article. I respect authors' right to do not respond to some minor recommendations.

Reviewer #4: Thank you for your revisions and incorporating my comments. However, please address the following comment in the discussion section and please refer to my previous comment in the previous revision for context.

The discussion needs incorporate findings of other papers in the literature in greater detail. This includes comparing and contrasting the authors' findings with other studies which have contrasting results.

For example, in lines 272 of tracked version, the authors note that “handwashing after removing feces was not always mentioned and surfaces were inconsistently cleaned with soap or disinfectants”. This is an important point to address considering other studies note that handwashing can lower infant exposure to pathogens. Other points that the authors could discuss include their results from Table 1 such as latrine features, floor material etc. Studies in the literature have identified these features to be associated with infant pathogen exposure.

These are just a few examples of how the authors can enhance their discussion. Please add additional examples of discussing results with other papers in the literature.

Reviewer #5: The authors are commended for having done a rigorous job in addressing comments made by reviewers. For reviewer comment R5.2, though the authors responded that animal health is outside their scope, they have responded accordingly in lines 37-38.

I have a few areas I have noted minor issues in wording as follows:

1. Line 211- Change "of" to "from"

2. Line 218: Change "child" to "child's"

3. Line 262: Remove the word "of"

4. Line 271: Change "animal" to "animals"

7. PLOS authors have the option to publish the peer review history of their article (what does this mean?). If published, this will include your full peer review and any attached files.

**Do you want your identity to be public for this peer review?** For information about this choice, including consent withdrawal, please see our Privacy Policy.

Reviewer #3: **Yes: **Joel Nkiama N. KONDE

Reviewer #4: No

Reviewer #5: No

---

## [Editor Report · Decision Letter 2]

27 Aug 2024

Multilevel factors drive child exposure to enteric pathogens in animal feces: A qualitative study in northwestern coastal Ecuador

PGPH-D-24-00231R2

Dear Dr.Karen Levy,

We are pleased to inform you that your manuscript 'Multilevel factors drive child exposure to enteric pathogens in animal feces: A qualitative study in northwestern coastal Ecuador' has been provisionally accepted for publication in PLOS Global Public Health.

Best regards,

Muhammad Asaduzzaman, MD MPH MPhil

Academic Editor
